# The Influence of Collusive Information Dissemination on Bidder's Collusive Willingness in Urban Construction Projects

**Xiaowei Wang [1], Kunhui Ye [1,2], Taozhi Zhuang [1,\* and Rui Liu [3]**

[1] School of Management Science and Real Estate, Chongqing University, Chongqing 400045, China; wangxiaowei313@cqu.edu.cn

[2] Research Center for Construction Economy and Management, Chongqing University, Chongqing 400044, China; Kunhui_YE@cqu.edu.cn

[3] Department of Construction and Real Estate, School of Civil Engineering, Southeast University, Nanjing 211189, China; 230218718@seu.edu.cn

\* Correspondence: tz.zhuang@cqu.edu.cn

**Abstract:** The process of urbanization and urban regeneration is inseparable from the implementation of urban construction projects. Current studies show a large amount of collusive bidding in urban construction projects, which has seriously affected healthy and sustainable urban development. Therefore, the governance of collusive bidding in urban construction projects is crucial to sustainable urbanization and urban renewal. In reality, the collusion information dissemination (CID) is a key influential factor in the bidder's collusive willingness (BCW). Knowing the influence of CID on BCW will help city managers to have a clearer understanding of the causes and governance focus of collusive bidding. Thus, the study using the multi-agent simulation technology simulates the influence of CID on BCW in different market scales, communication intensities, and trust boundaries based on the Deffuant model. The research found that the negative impact of the CID on the market is more incredible in cities with smaller market sizes, and effectively inhibiting the CID can reduce the occurrence of collusion. Moreover, the research also found that colluders always form their collusive alliances within CID networks. These findings suggest that urban managers should strengthen the suppression of collusive bidding by weakening the dissemination of collusive information and blocking the CID networks.

**Keywords:** collusive bidding; Deffuant model; multi-agent simulation; countermeasures

## 1. Introduction

The world population has grown significantly and our economies have become more industrialized over the past few hundred years, and, as a result, many more people have moved into cities [1,2]. This process is known as urbanization [2,3]. As shown in the statistic, the degree of urbanization in China, the world's second-largest economy, rose from 36 percent in 2000 to around 63.89 percent in 2020 [2]. With the development of urbanization and industrial upgrading, urban renewal is also becoming more common. The accelerated urbanization and large-scale urban regeneration mean that the whole of society has benefited from the construction industry. However, the current research found that a great deal of questions have been generated by urban construction [4]. For example, Owusu et al. [5] found that severe corruption behavior in the urban construction process weakened urban environmental management and increased urban susceptibility to extreme impacts of natural and human-made disasters. Wang et al. [3] demonstrated that collusive bidding is the most severe and illegal behavior in urban construction projects. Widespread collusive bidding has been identified as the primary inhibitor to the health and sustainability of urban development [3].

Collusive bidding refers to cases in which independent firms disclose their bidding prices to each other before the bidding process starts [6]. This practice allows the bidding firms to predetermine who will win the contract [7], which creates a non-competitive bidding environment, increases standard market costs, and causes economic damage to non-cartel bidders [8–11]. Projects acquired by collusive bidding always have serious safety risks and quality problems threatening public safety [3,12]. In urban cities, collusive bidding accounts for ill conditions ranging from undermining the city's construction market integrity system, impairing city building performance, and increasing city maintenance costs [1]. Therefore, the governance of collusive bidding is vital to sustainable urbanization and urban renewal.

Previous studies have provided a more serious insight into collusive bidding in the same area [7]. For example, Wang et al. [13] empirically analyzed the influence of the external environment on collusive bidding and found that bidders' collusive willingness and collusive scale are higher in the same province. Price [14] stated that there would be more business contacts between firms in the same area, thereby increasing the collusive bidding. The above results may suggest that the collusion information spreads more easily in the same city [15]. In reality, with the collusion information dissemination (CID), bidders find that many companies around them have conspired to obtain projects, but they are not discovered. This increases the bidder's trust in the current environment, where collusion is not be found, thus generating a fluke and colluding action. From this point of view, we can find that the CID is an important reason for the deterioration of collusive bidding in urban construction projects.

However, in the area of collusive bidding studies, there is little research on the relationship between CID and bidder's collusive willingness (BCW) in urban construction projects [16]. Revealing the impact of CID on BCW can not only provide a new research basis for subsequent collusion research, but also make city governors clearer about the negative impact of CID on cities. Therefore, this study, using multi-agent simulation technology, has simulated the influence of CID on BCW based on the Deffuant model. The results show that CID positively impacts BCW, and that, the smaller the city market, the more severe the effect. Moreover, the results also display that collusive bidding is more likely to occur between firms in some small collusive groups in the CID network. These results provide a valuable basis for urban managers to formulate collusive governance strategies.

## 2. Literature Review

### 2.1. Collusive Bidding in Urban Construction Projects

In reality, the process of collusive bidding is very complicated [17]. Firstly, the leading colluder may evaluate all parties' interests and calculate the possible collusion cost [18]. Secondly, according to the collusion cost that it can bear, the leading colluder contacts the bidders to persuade them to participate in their collusion organization, negotiate the collusion remuneration, and form a collusion team [19]. Finally, the leading colluder manipulates all collusive members' bids and decides that he is the final winning bidder by default [6]. Current research on collusive bidding mainly focuses on the collusive reason [12,20–24], determining factors [12,21,25–28], penalty [21,24–26,28–30], influence factors [7,20,21,25,26,31,32], governance [21,33,34], collusion forms [25,26,31,35–37], and collusion cost [30,38].

Collusive bidding has become a hot issue in the study of city governance and has attracted widespread attention in the researchers. For example, Shan et al. [39] used an artificial neural network method to assess collusion risks in managing construction projects. Signor et al. [40] utilized statistics and probability to identify and control collusion in public- and private-sector tendering in infrastructure projects. Wang et al. [13] showed that the higher the willingness to collude, the higher the probability of colluding. Owusu et al. [1] tackled corruption in urban infrastructure procurement, dynamic evaluation of the critical constructs, and anticorruption measures. Owusu et al. [41] also exposed the

impacts of anticorruption barriers on the efficacy of anticorruption measures in infrastructure projects. They showed that corruption is more likely to occur in urban infrastructure projects. Moreover, Wang et al. [3] disclosed the influence paths of the urban environment on bidders' collusive willingness and found that the number of collusive bidders is the most critical transmission medium in this path. Wang et al. [30] and Zhang et al. [24] also revealed that collusion among bidders is the most common form of collusion. It can be seen that the research on collusive bidding in the field of urban governance mainly focuses on collusion risk assessment, collusion identification, collusion governance measures, and other forms of collusion.

### 2.2. Influencing Factors of Collusive Bidding

Existing research shows that numerous factors influence the formation of collusive bidding in projects [42]. From an external environment perspective, Shi et al. [43] found that the main factors that encourage collusive behavior in bidding are the excessive competition and low profit margins experienced by contractors in the construction industry, making collusive bidding easy to encounter in a market with few competitors [44]. In reality, larger companies are more likely to submit larger offers and seek higher profits [6], and the larger market share of large companies enables them to implement collusive strategies to maintain their competitive advantage. Bolotova et al. [45] verified the above statement and stated that, if the external environment can promote them to obtain more profits, the collusive bidding is easier to achieve. In addition, the construction market structure, which consists of a small number of large firms and an overwhelming majority of small and medium firms, is an important factor contributing to collusive behavior in project bidding [44]. Market conditions such as the number of competitors, barriers to entry, frequency of interactions, market transparency, demand growth and volatility, business cycles, market share distribution, and cost asymmetries also greatly influence the decision to implement a collusive scheme [46].

From a contractor's perspective, Dorée [20] took the Dutch construction industry as an example to show that the greed of contractors is the main reason for the collusive bidding in construction. Zarkada-Fraser [42] showed that the contractor's affectivity and cognition can also influence the collusive bidding decisions. From the project perspective, Ratshisusu [47] states that the project scale will lead to the occurrence of collusive bidding. From the legal perspective, the imposition of administrative penalties on bid rigging has long been a common tool used by governments to warn bidders to respect free competition [12,48,49]. For example, Wang et al. [18] and Oke et al. [12] agree that the main reason for collusive bidding is the lack of punishment and supervision. In practice, France fined 21 construction companies EUR 17.3 million for collusive bidding in motorway projects [12]. The Netherlands penalized 344 companies for bid rigging in public works contracts linked to high-speed rail in Belgium and France [12]. The emerging cases of collusion are enough to show that the existing administrative penalties are still unable to prevent the occurrence of collusive bidding in urban construction projects [48]. The above research show that collusive bidding is caused by the comprehensive influence of multiple factors such as the external environment, project factors, the contractor's characteristics, and the law.

### 2.3. Collusive Networks in Bidding

Current research on the collusive networks in construction projects by Reeves-Latour and Morselli [16] verified the network patterns that underlie the making and sustainment of bid-rigging and found that participants with a higher degree of centrality were more likely to both be found guilty and receive greater sentences. They concluded that being in the thick of such conspiracies rendered an actor vulnerable, a proposition that has been consistent across crime network research [50]. Carlo and Marie [6] focused on collusion's network similarity in construction and emphasized the need to develop a monitoring system that allows researchers and analysts to track collusion patterns in various ways so as

to prevent an increase in more sophisticated schemes and cartels. Bunt [51] supposed that collusive members usually participate fully in other social groups and networks, and that it is this social embeddedness that increases the chances of maintaining secrecy. In addition, Xiao et al. [52] used the social network method to examine the collusive relationships in the Chinese construction industry, and found that the proposed social network model of deliberating bid riggers' relationships lays a solid foundation for the detection of collusive bidding in the construction sector. These studies indicate that, even amidst complex schemes by collusive firms to coordinate activities and outcomes, and even while maintaining an appearance of bidding competitiveness, these firms operate in a closed system that sets them apart from non-cartel competitors. Collusive firms, therefore, become overly similar in their bidding patterns.

## 3. Methodology

This study aims to reveal the influence of CID on BCW and proposes governance countermeasures for collusive bidding in urban construction projects. The following five steps were used to achieve the goals (Figure 1): First, the study raises research questions by analyzing existing literature and collusion cases. Second, the Deffuant model was selected for this research method, and the simulation model for the impact of CID on BCW were set based on the rules of CID and the collusive relationship network [53]. Third, the study designed the simulation experiment according to the collusive cases in Chinese construction [54] and in existing literature [55]. Fourth, the influence of CID on BCW in different market sizes, communication intensities, and trust boundaries were simulated using multi-agent simulation technology. Finally, the study put forward collusive governance countermeasures by the simulation result.

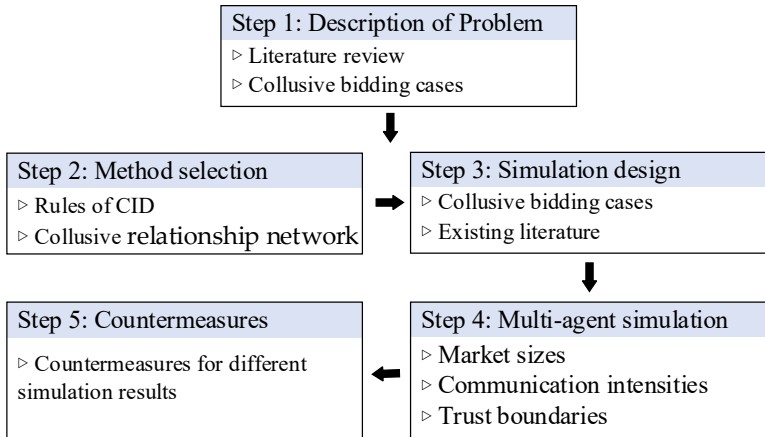

**Figure 1.** The framework of the methodology.

### 3.1. Simulation Rules Setting

Collusion relationship network is formed through the interaction of collusion information among the firms in the cities. Collusion information is continuously propagated in the network to influence bidders' responses to collusive bidding decisions. The Deffuant model is the most widely used continuous dynamic model. It follows the principle of "trust boundary". That is, when two individuals interact, if the difference between their views does not exceed a given threshold (trust boundary), the two can interact with points of view [56]. In current study, the Deffuant model was used by Yan [57] to analyze the influence of information dissemination in the platform on investor behavior, which is in line with the purpose of this study. Therefore, the Deffuant model was used in current research to build the bidder's collusive trust interaction model. In the Deffuant model, each individual is determined by two variables, namely, the opinion value $x$ and the un-

certainty $u$. For two random individuals, $i$ and $j$, the opinion values are $x_i$ and $x_j$, respectively, the uncertainty is $u_i$ and $u_j$, respectively, and $h_{ij}$ is the overlapping part of the two opinions. If $h_{ij} > u_i$, the opinion value and uncertainty of individual $j$ after interacting with $i$ are updated as:

$$x_j{'}=x_j+\mu\left(\frac{h_{ij}}{u_i}-1\right)(x_i-x_j) \tag{1}$$

$$u_j{'}=u_j+\mu\left(\frac{h_{ij}}{u_i}-1\right)(u_i-u_j) \tag{2}$$

In Formulas (1) and (2), $\mu$ is a constant to control the speed of opinion evolution. If $h_{ij} < u_i$, the opinion of individual $i$ will not affect $j$.

Based on the Deffuant model, the research considers the importance of individuals in the influence network. It establishes a network model of colluders based on the complex network, namely the revised method by Yan [51]. Assuming that the market size is N, the individual level (agent), then the state space of an individual is $X = (x_i, u_i, d_i, o_i)$. Among them, $x_i$ represents the trust value of the bidder in the safe collusive bidding environment, and the value range is [–1, 1]. $x_i(t)$ represents the attitude of bidder $i$ at time $t$. $u_i$ represents the objective uncertainty of bidder to the security of collusive bidding environment; the value range is (0, 0.5). $d_i$ represents the degree size of bidder in the relationship network. $O_i$ represents the bidder's collusive decisions, $O_i = 1$ represents the bidder's choice of collusion, and $O_i = −1$ represents no choice of collusion. The new update rules are as follows:

$$x_j{'}=x_j+\frac{1}{n_i-1}\sum_{i=1}^{n_i}\frac{d_i}{d_{it}}\mu\left(\frac{h_{ij}}{u_i}-1\right)(x_i-x_j) \tag{3}$$

$$u_j{'}=u_j+\frac{1}{n_i-1}\sum_{i=1}^{n_i}\frac{d_i}{d_{it}}\mu\left(\frac{h_{ij}}{u_i}-1\right)(u_i-u_j) \tag{4}$$

There are two aspects affected by the collusive bidding:

(1) Bidder's environmental trust level: the bidder's trust value in the collusion environment is $x$, and the uncertainty is $u$. Therefore, the relationship between the two becomes an important basis for bidders to collude in decision-making. If $x + u > 0$, bidders will tend to collude; if $x + u \leq 0$, bidders will not choose collusion;

(2) Group level: as a person, bidders have a herd mentality, and when their trust in the external environment is low, they are easily affected by the behavior of surrounding bidders in the bidding process, and it is challenging to maintain independence. This process follows the interaction rules (3) and (4). Therefore, for bidder $i$, the behavioral decision based on collusive environmental safety has:

① When $x_i + u_i \leq 0$, bidders will not make collusive decisions, where $O_i = −1$.

② When $x_i + u_i > 0$, the bidder has a 30% probability of making a collusive decision, $O_i = 1$, and a 70% probability of not making a collusive decision, $O_i = −1$.

*3.2. Simulation Scenario Setting*

In practice, different market sizes have different numbers of firm [57]. Wang, et al. [7] found that the larger the market sizes, the greater the number of firms involved in collusive bidding. Thus, the market sizes may also affect CID's impact on BCW. Besides, the communication intensity represents the bidders' security perception of external environment in this study. The higher the security perception, the more bidders are willing to communicate the facts of collusion with others, meaning more bidders can participate in their collusion team to form collusive alliances. Therefore, the influence of CID on BCW in different market sizes and communication intensities are simulated using multi-agent simulation technology.

In the Deffuant model, when the bidder's trust domain of the bidding environment overlaps $h_{ij} > u_i$, the two bidders will interact with collusive information [57]. In the process of communication, bidders will form a state of bipolar confrontation in the "migration" with different views [55]. However, in a multi-network environment, the trust boundary between bidders is broken. Two bidders distrusting the bidding environment will also interact collusive information due to project profitability and their benefits, thus affecting other bidder's collusive behavior [52]. Therefore, the study simulates the influence of CID on BCW under trust boundary and no trust boundary.

*3.3. Simulation Parameter Setting*

Before the simulation model is constructed, we need to set the simulation parameters. First, existing collusive cases show that construction firms in various cities range from dozens to more than a thousand. According to the number of construction firms corresponding to these cases, this study sets the number of agents as N=200, 500, 1000 to represent different cities. Second, Zhu, et al. [15] found that the bidder network has features of a small world at macro levels. Because the external environment may affect the simulation result, the study chooses the small world network as the simulation environment. Third, according to the information interaction value and uncertainty value set in the research of Yan [57], the information interaction value and uncertainty value of this study are set at t = 0, the bidder's collusion information interaction value is X = N(0, 1), and the uncertainty value of whether the collusion will be detected is *u* = U(0, 0.5). Table 1 shows the results of the parameters of this experiment.

**Table 1.** Parameter setting in simulation experiment.

| Parameter | Definition | Value or Range |
|:---:|:---:|:---:|
| N | number of the enterprise | (200, 500, 1000) |
| SW | type of network | - |
| TB | trust boundary | Yes/No |
| X | collusion information interaction value | N(0, 1) |
| *u* | uncertainty value | U(0, 0.5) |
| μ | bidders' communication intensity | 0.2, 0.3, 0.4, 0.5, 0.6 |

*3.4. Multi-Agent Simulation Technology*

Multi-agent simulation technology can be used as predictive tools for spurring innovations in society and our daily lives [58]. This practice can represent individual decision-making in detail to reproduce the complex phenomena that arise from the outcome of interactions between different agents. Therefore, a multi-agent model can be used to analyze the influence of collusive information interaction among bidders on collusive decision-making. A multi-agent system is composed of a simulation environment, objects, agents, relations, and operations [59]. The simulation environment is usually a space, such as a company or a city. Objects refer to individuals who can make decision-making responses in the environment. Agents are specific objects and represent the active entities in the system. Relations link objects to one another. An assembly of operations makes it possible for the agents to perceive, produce, transform, and manipulate objects in a simulation environment. We can simulate the behavior of individuals in the environment by changing the above parameters. In this study, the simulation environments were set by different cities, bidders are objects and relations produced by CID. The software anylogic 8.2.4 was used to simulate the influence of CID on BCW in different market sizes, communication intensities, and trust boundaries.

## 4. Results and Discussion

### 4.1. Influence of CID on BCW under Different Market Sizes

Figure 2 shows the simulation results of the impact of CID on BCW under different market sizes. In Figure 2, the ordinate is the simulation time, and the abscissa is the number of colluding firms. The simulation results show that the development trend of BCW is almost similar under the three market sizes, first increasing and then leveling off. The smaller the market size, the sooner it will stabilize. Moreover, the results also show that a higher proportion of colluding bidders eventually reach a collusive equilibrium in the smaller market. This means that the negative impact of CID on the market is more significant in a smaller market. This result is in line with reality. In reality, when a city is smaller, a message spreads faster across the town, and more people will be affected by the news. CID presents the same spread rules as above. Therefore, urban governors should strengthen the governance of CID in small cities, thereby reducing the negative impact of collusive bidding on the city.

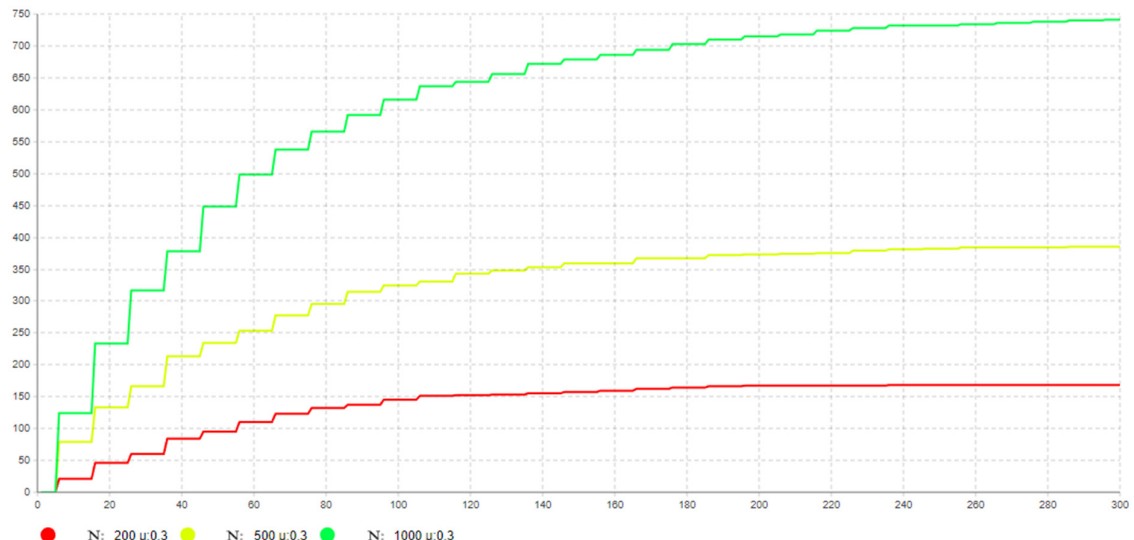

**Figure 2.** Influence of CID on BCW under different market sizes.4.2. Influence of CID on BCW under Different Communication Intensities

Figure 3a–c is the simulation results of the influence of CID on BCW under different communication intensities when the market size is 200, 500 and 1000. From Figure 3a–c, it can be seen that the more firms participate in the collusion, the faster the market environment reaches a state of collusive equilibrium with the increase in communication intensity. One reason for this is because, the more collusive firms involved, the more colluders can spread collusive information, which leads to an increase in collusive bidding in urban construction projects. Another reason is that the increase in communication intensity will make more bidders receive colluding information and lead to the dissemination of collusive details, thereby increasing the number of collusive bidding. The study also found that, when the communication intensity increases, the proportion of colluding bidders is higher in the equilibrium state. This is because the CID helps enterprises to establish collusive trust. Collusive trust increased the bidder's collusive behaviors. The above results show that effectively inhibiting the transmission of collusion information between firms can reduce the occurrence of collusion. Thus, the urban governors can encourage firms to reduce the transmission of collusion information through reporting, complaints, and rewards. These governance measures would be conducive to reducing the expansion of the collusion network and weakening the collusion behavior.

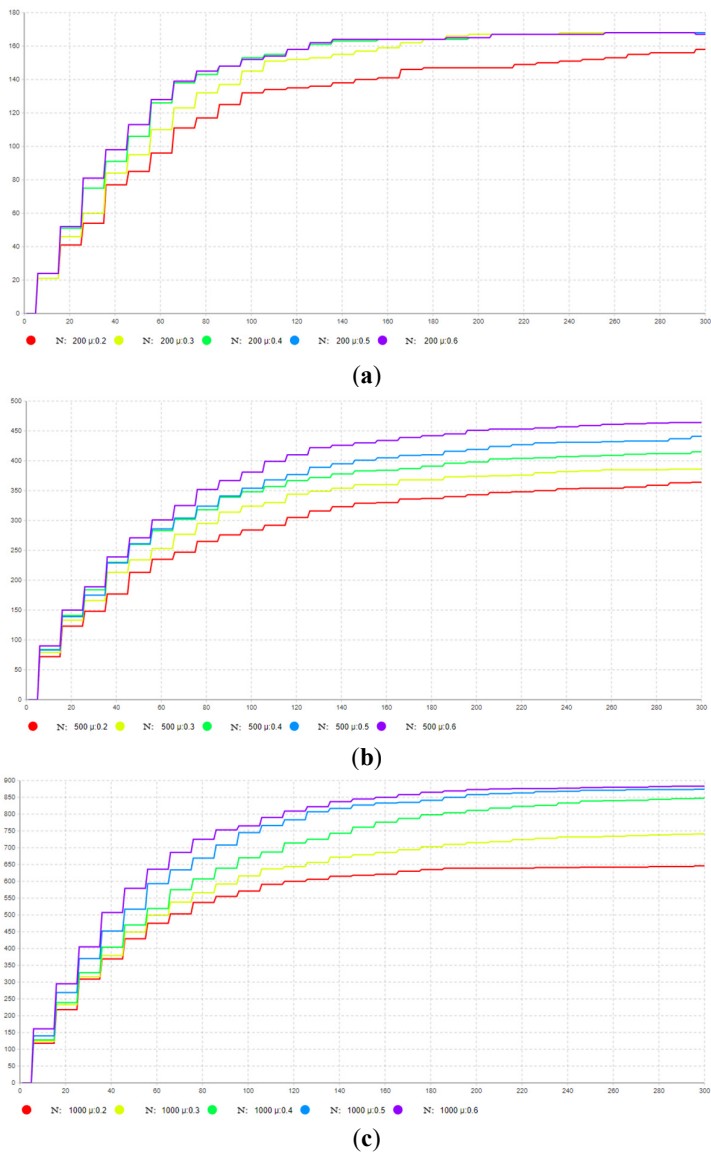

**Figure 3.** Influence of CID on BCW under different communication intensities.

*4.2. The Evolution of the Influence of CID on BCW under Trust Boundary and No Trust Boundary*

Figure 4 is a screenshot of the influence of CID on BCW under trust boundary and no trust boundary at t = 30 and 200 when the market size is 200. Figure 4a,b represents the evolution process of bidders' collusive decision-making in the market when there is a trust boundary. Figure 4c,d represents the evolution process of bidders' collusive decision-making in the market when there is no trust boundary. It can be seen from the figure that a large number of bidders change from a non-collusion state to a collusion state regardless of whether there is a trust boundary or not. From Figure 4a,b, it can be seen that the trust boundary can aggravate the "aggregation" of bidders, so that bidders can communicate in the collusive network to reach a collusive alliance, and, to some extent, become isolated from other colluding groups. When there is no trust boundary, bidders can receive more collusive information, and their behavior is more susceptible to the behavior of surrounding bidders. However, from the result of t = 200, the study found that the number of firms converted into collusion in the case of no trust boundary is less than the number of firms

converted into collusion in the case of a trust boundary. Although bidders can communicate more in the case of no trust boundary, collusion is illegal, and reaching a collusive agreement with bidders outside a small collusive group has risks. Therefore, it is easier for bidders to reach a collusive consensus with those whom they have a relationship with.

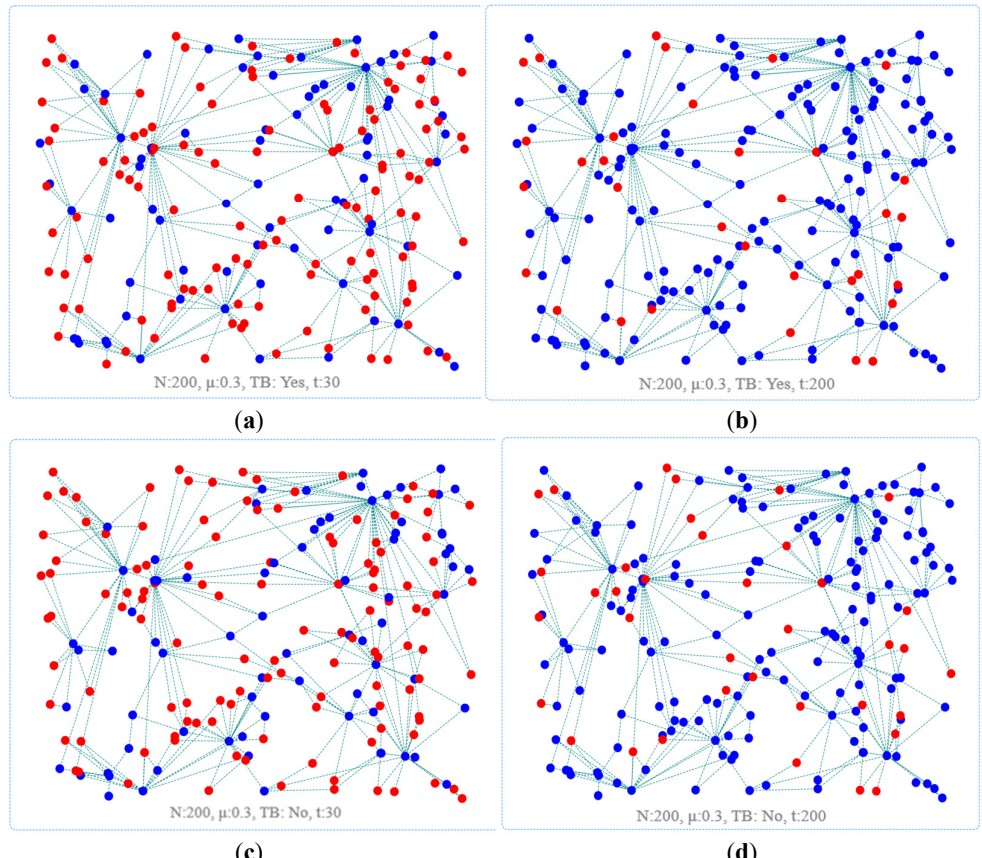

**Figure 4.** Influence of CID on BCW under trust boundary (Number of firms = 200). Note: the red node means normal bidders, the blue node means collusive bidders, and the node's size represents the degree of the node.

Figure 5 is a screenshot of the evolution of the influence of CID on BCW with and without trust boundaries when the market size is 500 at t = 30 and 200. Compared with the case where the market size is 200, when the market size is 500, the number of various small groups formed in the market is greater, and the relationship network connected by each group is more complex. Therefore, more companies can be contacted within a small group to form a collusive bidding team. The results validate Wang et al.'s [13] findings from different perspectives. In this case, the competition is more intense, and the number of colluding groups in the same project increases [7]. Meanwhile, there may be multiple collusion groups to competing for the same project. From this, the critical crackdown on the relationship network in the larger market has become a significant focus of collusive bidding governance.

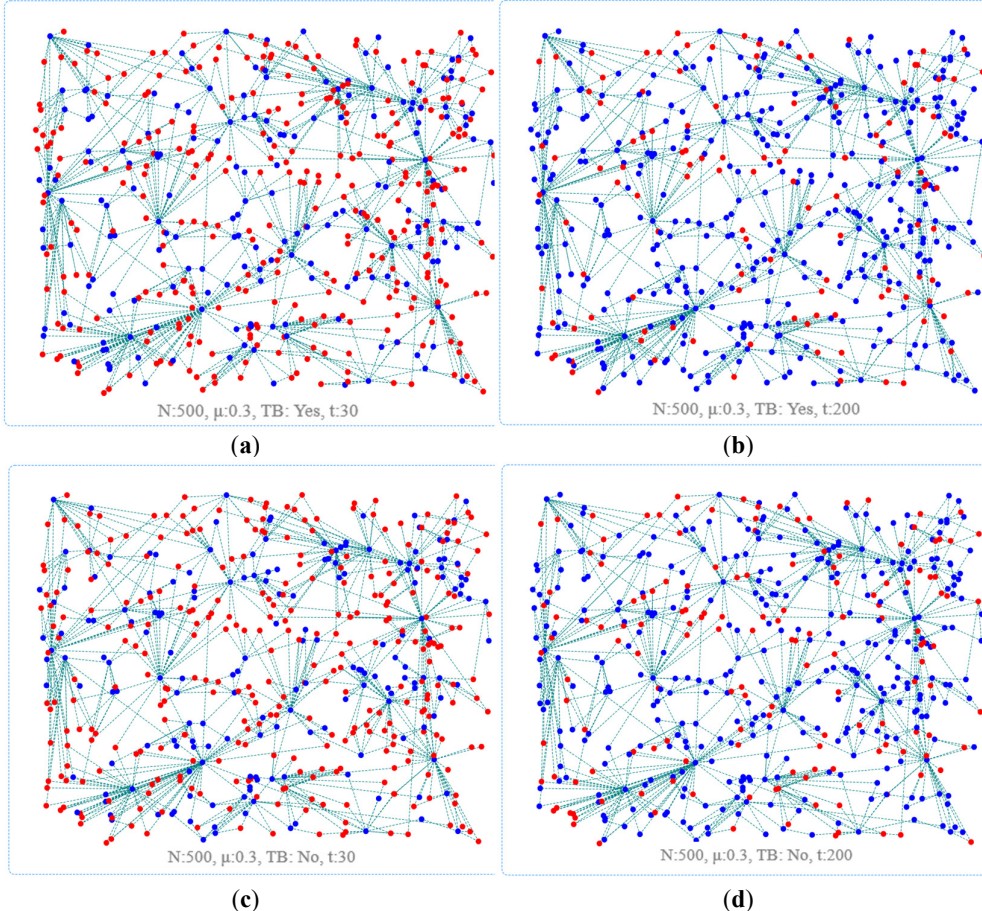

**Figure 5.** Influence of CID on BCW under trust boundary (number of firms = 500). Note: the red node means normal bidders, the blue node means collusive bidders, and the node's size represents the degree of the node.

Figure 6 shows the screenshot of the evolution of influence of CID on BCW with and without trust boundaries when the market size is 1000 at t = 30 and 200. It can be seen that, compared with the networks with 200 and 500 firms, the network with 1000 firms has more complex relationships and more small groups. Although more firms have moved from the non-collusive state to the collusive state over time, there are still many non-collusive firms in the market. This result means that the larger the market size, the more difficult it is to reach a collusive equilibrium. In the current research, Anderson and Cau [60] showed that implicit collusion often occurs in a relatively unstable way without a single equilibrium strategy being reached. The study reveals the collusion equilibrium from different perspectives, which is an extension of the above study. Moreover, compared with no trust boundaries, the conversion rate is faster, which is consistent with the analysis results of the market sizes in the state of 200 and 500. The collusive bidders contacted by the colluders, and the bidders that exchange collusive information are more than the bidders in the small group. This is because of mutual trust among the firms within the small group, which promotes the BCW. Thus, the most critical thing to govern collusive bidding in urban construction projects from the relational network perspective is to break the small collusive group among bidders and weaken the trust of collusive firms. Figure 6 also shows that each small group has a critical node. When we break the nodes, small groups are also damaged. Therefore, the study suggests that colluding networks can be defeated by focusing on attacking the main nodes in the network.

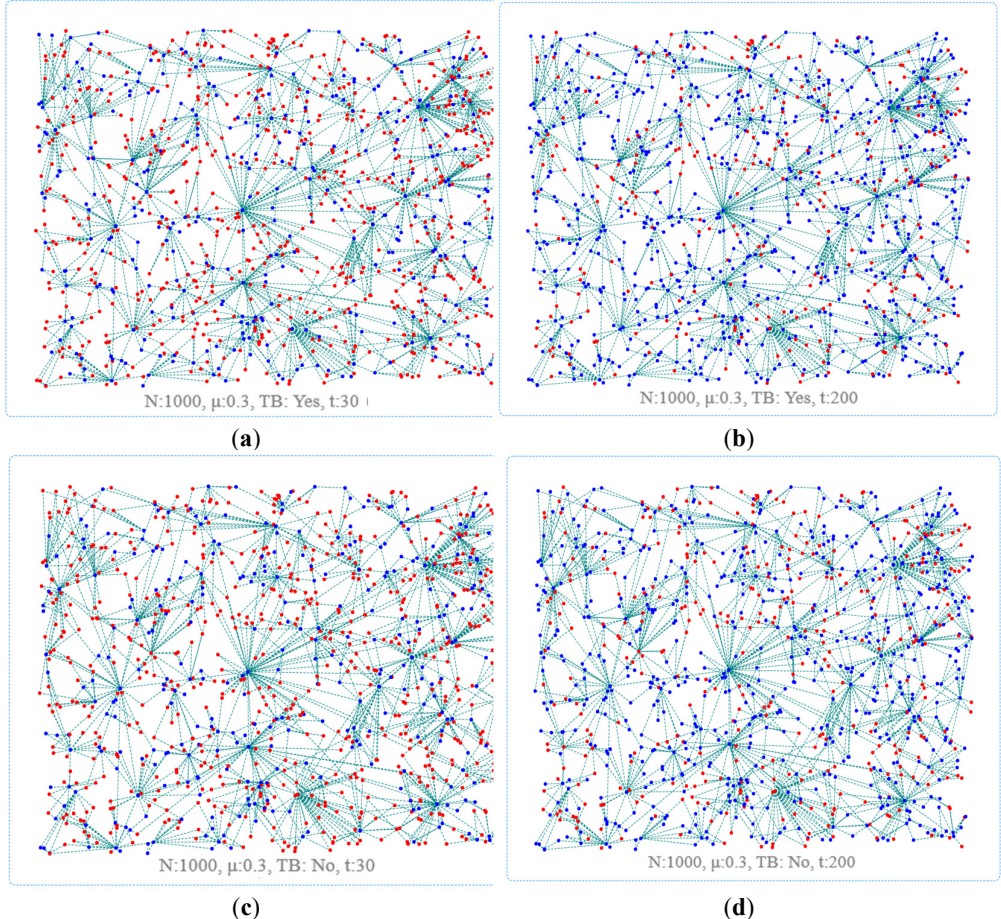

**Figure 6.** Influence of CID on BCW under trust boundary (number of firms = 1000). Note: the red node means normal bidders, the blue means collusive bidders, and the node's size represents the degree of the node.

## 5. Conclusions

The dissemination of collusive information in cities can make bidders have a fluke mentality of colluding and not being discovered, thereby increasing the willingness to collude. Exploring the impact of collusive information dissemination (CID) on bidder's collusive willingness (BCW) under different market scales, communication intensities, and trust boundaries will help city managers understand the causes of collusion deterioration to propose more effective collusion governance countermeasures. Therefore, this study uses multi-agent simulation technology to simulate the influence of CID on BCW. It is found that the negative impact of CID on the market is more severe in smaller market sizes. Moreover, with increased communication intensity, more bidders participate in the collusion, and the entire market environment reaches a state of collusive equilibrium earlier. Moreover, it is easier for bidders to reach a collusive team with those who they have a relationship with in collusive networks. When the market size increases, the key nodes in the collusive network become bigger, and the relationship network connected by each group is more complex.

This study reveals the impact of CID on BCW from a multi-dimensional perspective, which is an expansion of existing research on collusion networks. The study also suggests several implications for collusion governance. First, the results show that effectively inhibiting the transmission of collusion information between firms can reduce the occurrence of collusion, especially in cities with the smaller market size. Thus, the urban governors can encourage firms to reduce the transmission of collusion information through

reporting, complaints, and rewards. These governance measures will reduce the expansion of the collusion network and weaken the collusion behavior. Second, the research suggests that colluding networks can be defeated by focusing on attacking the key nodes in the network. The scathing attack on key nodes can effectively destroy the critical connection points in the collusion network and block the spread of collusion information. Third, the existing literature shows that the process of collusive bidding is not very different in different countries. Meanwhile, the decision-making process of bidders in various countries in organizing collusion is nearly the same. Therefore, the results of this study are also applicable to other countries.

Although this paper presents empirical findings, the study has two limitations: First, this study only considers the impact of different market sizes, communication intensities, and trust boundaries. The impact of other dimensions needs to be examined in the future. Second, based on the impact of CID on BCW, this study proposes some policy recommendations for collusive governance for city managers. However, these policy recommendations require further validation before use to ensure effective policy implementation.

**Author Contributions:** Conceptualization, K.Y.; methodology, X.W. and T.Z.; software, X.W. and T.Z.; validation, X.W. and R.L.; visualization, R.L.; writing—original draft preparation, X.W.; writing—review and editing, T.Z.; supervision, K.Y.; funding acquisition, K.Y. and T.Z. All authors have read and agreed to the published version of the manuscript.

**Funding:** This research was funded by National Natural Science Foundation of China (Grant No. 71871033), the MOE (Ministry of Education in China) Project of Humanities and Social Sciences (Project No. 21XJC630017), and the Fundamental Research Funds for the Central Universities (Grant No. 2021CDJSKJC19).

**Data Availability Statement:** The collusive cases are available at https://wenshu.court.gov.cn/ (accessed on 1 September 2021).

**Acknowledgments:** The authors sincerely thank the editors and the anonymous reviewers for their constructive suggestions for this manuscript.

**Conflicts of Interest:** The authors declare no conflict of interest.

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
