# Peer review of "The Influence of Collusive Information Dissemination on Bidder’s Collusive Willingness in Urban Construction Projects"

_land, doi:10.3390/land11050643_

Round 1
Reviewer 1 Report
Wang, X., et al. submitted an interesting study on The influence of collusive information dissemination on bidder’s collusive willingness in urban construction projects. This paper needs considerable work before it can be considered for publication. The authors are recommended to revise the paper based on the following comments.
General Comments:
- Keywords need to be more specific. Do not use the words you already mentioned in the title.
- This paper should be structured well and ensure a better flow of ideas.
- Insert more references where necessary
This paper can be considered for publication after addressing the following specific comments.
- Introduction
- What is novel in what you are doing? What is the scientific contribution of the paper? Please explain clearly.
- In-Page 2 - line 70: What is CBW? Did you mean BCW in here?
- How this research will help the city governors to manage the urban collusion?
- Literature Review
- You may give some starting remarks before starting section 2.1 and similarly some summarization at the end of the literature review section.
- Methodology
- The goal/objective of this research should be stated clearly at the beginning of the methodology.
- A methodological framework needs to be shown to indicate the flow of information. And the description needs to be aligned to the developed methodology framework.
- The data collection method and the source of data for this research have not been indicated in the menu script. This is a key section of this methodology. There is no significance in this research without proper data. I suggest authors give comprehensive information about the data collection, filtering and analysis method in the methodology.
- The methodology section needs to be structured well and written clearly. This is very important for the reader to understand the proposed research and to interpret your contribution to the existing knowledge.
- The methodology consists of some literature sections, which can be avoided to make the methodology section clear to the readers. You may use the literature in the literature review section instead of the methodology section. The methodology should be clear and comprehensive to the reader.
- Did you use any software to conduct the given simulations? If so what are those? And what are the assumptions that you assumed in your analysis?
- Results
- These results are pointless without the data sources and the assumptions you made in this analysis. Therefore, I do not have any idea about the validity of these results without a proper methodology. Please revise the methodology and then only we can check the results.
- The figures shown in this section need to be explained well in order to give a better idea to the readers.
- Did you validate these results? So what is the method that you used and how did you get data for the validation?
- Conclusions
- Discuss the generalizability and reproducibility of these research findings for similar applications in other countries.
Author Response
Point 1: General Comments. 1. Keywords need to be more specific. Do not use the words you already mentioned in the title. 2. This paper should be structured well and ensure a better flow of ideas. 3. Insert more references where necessary.
Response 1: Thanks for the comment.
- We changed the keywords from “urban construction projects; collusive information dissemination; collusive bidding; multi-agent simulation” to “collusive bidding; Deffuant model; multi-agent simulation; countermeasures”. This makes the keywords cover the research objects, theories, methods and countermeasures. Please see line 27.
- In the revised manuscript, we have improved both the content and structure of the study to make the structure and content clearer. For example, we have rewritten the introduction section to make the topic more prominent (lines 29-76). In addition, we adjusted the structure of the literature review section and classified and discussed some of the contents (lines 78-158). Moreover, significant revisions have been made to the methodology section (lines 159-172). In all, we have improved the entire manuscript. The modifications can find in the revised manuscript. More detailed responses can be find in the detailed responses in subsequent sections.
- Ok. We have added more references (in lines 46, 69, 79, 107, 164, 165 and so on) to the revised manuscript.
Point 2: Introduction. 1. What is novel in what you are doing? What is the scientific contribution of the paper? Please explain clearly. 2. In-Page 2 - line 70: What is CBW? Did you mean BCW in here? 3. How this research will help the city governors to manage the urban collusion?
Response 2: Thanks for the comment.
- We have rewritten the introduction section to better present the goals and contributions of this study. Please see lines 67-76. The following contents are this study’s goals and contributions. Existing research display that the dissemination of collusion information is an important reason for the deterioration of collusive bidding in urban construction projects. However, in the area of collusive bidding studies, there is little research on the relationship between CID and bidder’s collusive willingness (BCW) in urban construction projects. Revealing the impact of CID on BCW can not only provide a new research basis for subsequent collusion research but also make city governors clearer about the negative impact of CID on cities. Therefore, the study using the multi-agent simulation technology, simulates the influence of CID on BCW based on the Deffuant model. The results show that CID positively impacts BCW, and the smaller the city market, the more severe the effect. Besides, the results also display that collusive bidding is more likely between firms in some small collusive groups in the CID network. These results provide a valuable basis for urban managers to formulate collusive governance strategies.
- Yes. We changed the CBW to BCW.
- Revealing the impact of CID on BCW can not only provide a new research basis for subsequent collusion research but also make city governors clearer about the negative impact of CID on cities. We added the explanations in lines 69-71. Also, we changed the original sentence to “These results provide a valuable basis for urban managers to formulate collusive governance strategies.” Please see lines 75-76.
Point 3: Literature Review. 1. You may give some starting remarks before starting section 2.1 and similarly some summarization at the end of the literature review section.
Response 3: Thanks for the suggestion.
We made detailed revisions to the literature review section, especially sections 2.1 and 2.2 (lines 78-138). In section 2.1 (lines 78-104): First, we tell the readers that the process of collusive bidding is very complicated. Second, the manuscript describes in detail how the process of collusive bidding. This helps readers gain a better understanding of how collusive bidding is formed. Third, we discussed the main themes of existing collusion research. Finally, we described research on collusive bidding in urban governance and summarized representative works on the above themes. Besides, in section 2.2 (lines 105-138), we also reorganized the literature from the external environment, contractor, project, and legal perspectives. Moreover, we have added starting remarks and summarization in sections 2.1 and 2.2. For example, in lines 102-104, we added “it can be seen that the research on collusive bidding in the field of urban governance mainly focuses on collusion risk assessment, collusion identification, collusion governance measures and forms of collusion.” as summarization at the end. In lines 102-104, we added “Existing research shows that numerous factors influence the formation of collusive bidding in projects.” as starting remarks. In lines 135-138, we added “The above research show that collusive bidding is caused by the comprehensive influence of multiple factors such as the external environment, project factors, the contractor's characteristics, and the law.” as summarization at the end.
Point 4: Methodology. 1. The goal/objective of this research should be stated clearly at the beginning of the methodology. 2. A methodological framework needs to be shown to indicate the flow of information. And the description needs to be aligned to the developed methodology framework. 3. The data collection method and the source of data for this research have not been indicated in the menu script. This is a key section of this methodology. There is no significance in this research without proper data. I suggest authors give comprehensive information about the data collection, filtering and analysis method in the methodology. 4. The methodology section needs to be structured well and written clearly. This is very important for the reader to understand the proposed research and to interpret your contribution to the existing knowledge. 5. The methodology consists of some literature sections, which can be avoided to make the methodology section clear to the readers. You may use the literature in the literature review section instead of the methodology section. The methodology should be clear and comprehensive to the reader. 6. Did you use any software to conduct the given simulations? If so what are those? And what are the assumptions that you assumed in your analysis?
Response 4: Thanks for these valuable comments.
- Right. In lines 160-161, we added the goal of this research: “This study aims to reveal the influence of CID on BCW and propose governance countermeasures for collusive bidding in urban construction projects.”
- Ok. We have further revised the methodological framework of the original manuscript to make it clearer. For the revised content, please see the revised manuscript (lines 160-171). Besides, we also added a figure to show the framework of the methodology.
- In lines 241-255, we described the source of the simulation data. First, existing collusive cases show that construction companies in various cities range from dozens to more than a thousand. According to the number of construction companies corresponding to these cases, the study sets the number of agents as N=200, 500, 1000 to represent different market sizes. Second, Zhu, et al. [16] display that the collusive bidders’ network has features as a small world at macro levels. Thus, the study chooses the small world network as the simulation environment. Third, according to the information interaction value and uncertainty value set in the research of Yan [57], the information interaction value and uncertainty value of this study are set at t=0, the bidder’s collusion information interaction value is X =N(0, 1) and the uncertainty value of whether the collusion will be detected is u =U(0, 0.5). In this article, part of the simulation data is set through collusive bidding cases by Chinese construction industry, and part of the data is set from existing literature. The selection of data is based on the requirements of the Deffuant model. Many kinds of literature use similar method to set the corresponding data. For example, Huang, C. , et al. "Modeling of agent-based complex network under cyber-violence." Physica A Statistical Mechanics & Its Applications 458 (2016):399-411. Based on these papers, we believe the findings of this study are scientifically valid.
- Thanks for this suggestion. In the revised manuscript, we have made in-depth improvements to the content of the methodology section. This study's methodology includes four aspects. First, summarize the study's research framework. Second, the theoretical model and simulation rules in this study are determined. Third, the simulation experiment is designed, including scenario design and parameter design. Finally, the simulation technique used in this article is described. This makes the methodology of this study clear to the readers. Please see lines 159-272 in the revised manuscript.
- We have shortened the literature in the methodology section. To make the article more based on previous research, we have reserved some literature in the methodology section.
- Yes. This study uses the software of Anylogic 8.2.4 to simulate the impact of CID on BCW. We added this sentence to lines 271-272.
Point 5: Results. 1. These results are pointless without the data sources and the assumptions you made in this analysis. Therefore, I do not have any idea about the validity of these results without a proper methodology. Please revise the methodology and then only we can check the results. 2. The figures shown in this section need to be explained well in order to give a better idea to the readers. 3. Did you validate these results? So what is the method that you used and how did you get data for the validation?
Response 5: Thanks for this review
- In the lines 159-272, the manuscript describes the methodology used in this study and how it was applied and the basis for selection. In the lines 241-255, the manuscript describes the source of the simulation data. Part of the simulation data is set through collusive bidding cases by Chinese construction industry, and part of the data is set from existing literature. In simulation research, a large body of literature adopts similar methods. In addition, to make the simulation environment more in line with the actual situation of the construction industry, this study sets the simulation environment based on the actual situation of the cities in China. Please see respond 4 for more explanation. Moreover, because this research mainly studies the impact of CID on BCW, the manuscript raises this question through literature review in the introduction (lines 55-66). Besides, in sub-section 3.2 “Simulation scenario setting”, the research explained the reasons for choosing different simulation scenarios (lines 222-239). These contents are sufficient to support the validity of this study. Therefore, setting assumptions is not of great necessity in this manuscript.
- Ok. In the revised manuscript, we have added more explanations and discussion of figures. In lines 281-283, we added “In reality, when a city is smaller, the faster a message spreads to the town, the more people will be affected by the news. CID presents the same spread rules as above.” In lines 295-299, we added “The one reason is the more collusive firms involved, the more colluders can spread collusive information, which leads to an increase in collusive bidding in urban construction projects. Another reason is the increase of communication intensity will make more bidders receive colluding information and lead to the dissemination of collusive details, thereby increasing the number of collusive bidding.” In lines 300-302, we added “This is because the CID helps enterprises to establish collusive trust. Collusive trust increased the bidder's collusive behaviors.” In lines 366-369, we added “In the current research, Anderson and Cau [60] show that implicit collusion often occurs in a relatively unstable way without a single equilibrium strategy being reached. The study reveals collusion equilibrium from different perspectives, which is an extension of the above studies.” For more explanations, please see the results and discussion section of the revised manuscript.
- Yes. In the results and discussion section, we discuss and verify the results using previous research findings and existing collusive practice. For example, In lines 347-350, the sentences “The results validate Wang et.al [13]’s findings from different perspectives. In this case, the competition is more intense, and the number of colluding groups in the same project will increase [7]. Meanwhile, there may be multiple collusion groups to competition the same project.” is used to verify the results in this article. Finally, collusive practice and existing research infer that the results conform to the laws of reality.
Point 6: Conclusions. 1. Discuss the generalizability and reproducibility of these research findings for similar applications in other countries.
Response 6: Thanks for this suggestion. The existing literature shows that the process of collusive bidding is not very different in different countries. Meanwhile, the decision-making process of bidders in various countries in organizing collusion is nearly the same. Therefore, the results of this study are also applicable to other countries. We added this discussion to lines 423-426.

Reviewer 2 Report
The influence of collusive information dissemination on bidder’s collusive willingness in urban construction projects is an interesting read.
The article contains a thorough review of the issue that is very important for the construction industry and beyond. The number of sources studied and cited in chapter 2 is adequate. However, in the future, I suggest reading the works of R. Porter and A. Foremny. Maybe authors will find additional ideas there.
For example:
Bidding rings and the winner's curse, K Hendricks, R Porter, G Tan, The RAND Journal of Economics 39 (4), 1018-1041
Collusion, RH Porter, JD Zona, Antitrust Law Journal 2, 1069
Collusion in industrial economics: a comment, RH Porter, Journal of Industry, Competition and Trade 5 (3), 231-234
Comparison of ANN classifier to the neuro-fuzzy system for collusion detection in the tender procedures of road construction sector, H Anysz, A Foremny, J Kulejewski, IOP Conference Series: Materials Science and Engineering 471 (11), 112064
Estimating potential losses of the client in public procurement in case of collusion utilizing a MLP neural networks, H Anysz, A Foremny, J Kulejewski, Czasopismo Techniczne 2014 (Budownictwo Zeszyt 1-B (5) 2014), 105-118
Review of collusion and bid rigging detection methods in the construction industry, A Foremny, Creative Construction Conference 2018, 946-953
The four-step methodology suggested by the authors is well thought out and described.
The results are interesting, proving that effectively inhibiting the transmission of collusion information between firms could reduce the occurrence of collusion. Authors also claim that colluding networks can be defeated by focusing on attacking the leaders in the network. It seems to be true. However, it would be good to perform additional studies checking the effectiveness of this approach in the construction practice.
The authors are aware of the limitations of the research and presented possibilities for further studies. It would be interesting to read about it in the future.
The language of the publication is correct. The publication reads well, it is interesting. The structure is mostly good. Conclusions are consistent with the evidence and arguments presented in the article.
Author Response
Point 1: The influence of collusive information dissemination on bidder’s collusive willingness in urban construction projects is an interesting read. The article contains a thorough review of the issue that is very important for the construction industry and beyond. The number of sources studied and cited in chapter 2 is adequate. However, in the future, I suggest reading the works of R. Porter and A. Foremny. Maybe authors will find additional ideas there. For example: Bidding rings and the winner's curse, K Hendricks, R Porter, G Tan, The RAND Journal of Economics 39 (4), 1018-1041; Collusion, RH Porter, JD Zona, Antitrust Law Journal 2, 1069; Collusion in industrial economics: a comment, RH Porter, Journal of Industry, Competition and Trade 5 (3), 231-234; Comparison of ANN classifier to the neuro-fuzzy system for collusion detection in the tender procedures of road construction sector, H Anysz, A Foremny, J Kulejewski, IOP Conference Series: Materials Science and Engineering 471 (11), 112064; Estimating potential losses of the client in public procurement in case of collusion utilizing a MLP neural networks, H Anysz, A Foremny, J Kulejewski, Czasopismo Techniczne 2014 (Budownictwo Zeszyt 1-B (5) 2014), 105-118; Review of collusion and bid rigging detection methods in the construction industry, A Foremny, Creative Construction Conference 2018, 946-953.
Response 1: Thanks for the comment and suggestion. I have downloaded and studied the articles and added two articles as citations in the revised manuscript (line 50 and line 79).
Point 2: The four-step methodology suggested by the authors is well thought out and described.
Response 2: Yes, thanks.
Point 3: The results are interesting, proving that effectively inhibiting the transmission of collusion information between firms could reduce the occurrence of collusion. Authors also claim that colluding networks can be defeated by focusing on attacking the leaders in the network. It seems to be true. However, it would be good to perform additional studies checking the effectiveness of this approach in the construction practice.
Response 3: Right. This study's significance is to find out the influence mechanism of collusion information dissemination on bidders' collusive willingness through simulation. The study reveals the effect of collusion information dissemination on bidders' collusive willingness under different market sizes, communication intensities, and trust boundaries. Meanwhile, the study put forward some recommendations for governance countermeasures based on these impacts. These recommendations do require further validation before they can be adopted. The study describes this limitation in the conclusion section. Please see lines 430-433.
Point 4: The authors are aware of the limitations of the research and presented possibilities for further studies. It would be interesting to read about it in the future.
Response 4: Thanks to reviewer 2 for acknowledging this study.
Point 5: The language of the publication is correct. The publication reads well, it is interesting. The structure is mostly good. Conclusions are consistent with the evidence and arguments presented in the article.
Response 5: Thanks again to reviewer 2 for acknowledging this study.

Round 2
Reviewer 1 Report
The authors have done a significant improvement towards the comments and suggestions that were suggested in my previous report. I believe the present form of the article can be considered for publication in the journal.